# Current Advances in Biodegradation of Polyolefins

**DOI:** 10.3390/microorganisms10081537

**Published:** 2022-07-29

**Authors:** Ni Zhang, Mingzhu Ding, Yingjin Yuan

**Affiliations:** 1Frontier Science Center for Synthetic Biology and Key Laboratory of Systems Bioengineering (Ministry of Education), School of Chemical Engineering and Technology, Tianjin University, Tianjin 300072, China; zhangn07@tju.edu.cn (N.Z.); yjyuan@tju.edu.cn (Y.Y.); 2Collaborative Innovation Center of Chemical Science and Engineering (Tianjin), Tianjin University, Tianjin 300072, China

**Keywords:** polyolefins, polyethylene, biodegradation, artificial microbial consortia

## Abstract

Polyolefins, including polyethylene (PE), polypropylene (PP) and polystyrene (PS), are widely used plastics in our daily life. The excessive use of plastics and improper handling methods cause considerable pollution in the environment, as well as waste of energy. The biodegradation of polyolefins seems to be an environmentally friendly and low-energy consumption method for plastics degradation. Many strains that could degrade polyolefins have been isolated from the environment. Some enzymes have also been identified with the function of polyolefin degradation. With the development of synthetic biology and metabolic engineering strategies, engineered strains could be used to degrade plastics. This review summarizes the current advances in polyolefin degradation, including isolated and engineered strains, enzymes and related pathways. Furthermore, a novel strategy for polyolefin degradation by artificial microbial consortia is proposed, which would be helpful for the efficient degradation of polyolefin.

## 1. Introduction

Plastics are one of the most widely used and vital materials in the modern world. However, the enormous manufacture and abuse of plastics place a huge burden on the environment. It was reported that about 58% of plastic waste was placed in landfills or discharged directly into the environment, 24% was burned and only 18% of plastic waste was recycled globally [1]. Some researchers have estimated that there would be roughly 12,000 Mt of plastic waste in the natural environment by 2050 [2]. Petroleum-derived polymers (polyethylene (PE), polyethylene terephthalate (PET), polyurethane (PU), polystyrene (PS), polypropylene (PP) and polyvinyl chloride (PVC)), especially polyolefins (PE, PP and PS) are widely used for packaging materials, plastic carry bags, plastic films, packaging foam, disposable cups and food containers [3] and are extremely recalcitrant to natural biodegradation [4]. As shown in Figure 1, polyolefins are linked by C–C and C–H bonds, the bond energy of which is much higher than that of C–O and C–N bonds, which means that polyolefins are more recalcitrant to degradation than plastics that consist of ester bonds, such as PET and PU [5]. There are different types of PE, including HDPE (high-density polyethylene), LDPE (low-density polyethylene) and LLDPE (liner low-density polyethylene). It is more difficult to degrade HDPE than LDPE or LLDPE due to its high density and crystallinity [1,2]. Moreover, because of its branches, PP is more difficult to degrade than PE. The benzene ring in PS made it more difficult to degrade than PE and PP. Although chemical treatment and thermal pyrolysis have been reported for polyolefin degradation, the use of harsh reaction conditions and toxic organic solvents can lead to secondary pollution in the environment. In recent years, many types of PE- and PS-degrading living species (mainly microbes) have been identified. These findings provide us with a mild and ecofriendly method for the degradation of polyolefins. Although the biodegradation efficiency of polyolefin is not high, polyolefin biodegradation is a promising method of recycling polyolefins, since various biotechnologies are being developed rapidly [5]. 

It was reported that the degradation rate of PP and PS was improved after pretreatment by UV and high temperature [6]. Therefore, many researchers have pretreated plastics with UV or high temperature prior to biodegradation. Some microorganisms, including pure strains and microbial consortia, have been isolated from the environment for the degradation of polyolefins. In 2006, Sivan et al. [7] isolated *Rhodococcus ruber* C208, which adhered to PE films and utilized LDPE films as the sole carbon source. Later, researchers found that *R. ruber* C208 could secrete laccase in vitro and that laccase also play a vital role in PE degradation [8]. In addition to laccase, manganese peroxidase [9], soybean peroxidase [10] and alkane hydroxylase [11,12] are also capable of degrading PE. Few PP-degrading enzymes have been reported to date. Hydroquinone peroxidase, which is produced by *Azotobacter beijerinckii* HM121, was reported to degrade PS [13]. Moreover, many microbial consortia have been isolated from nature with the capacity to degrade polyolefins. Some microbial consortia can degrade polyolefins into long-chain aliphatic compounds, which include alkanes, alkenes, alcohols and acids [14,15]. Many alkane hydroxylases that can degrade aliphatic alkanes have been identified, as well as alkene mono-oxygenases, which can degrade short-chain alkenes [16,17,18,19,20]. These researchers have provided a basis for the construction of artificial microbial consortia for the highly efficient degradation of polyolefins.

This review summarizes natural strains, enzymes and engineered microbial chassis for the biodegradation of polyolefins. Based on the former content, we provide an idea for polyolefin degradation by artificial microbial consortia and discussed the prospect.

## 2. Biodegradation of Polyolefins

Polyolefins, including PE, PP and PS, are composed of C–C and C–H bonds, which are more stable against degradation than ester bonds. During polyolefin biodegradation, C–C and C–H bonds are oxidized. Many types of microorganisms have been isolated from sea water, compost and activated sludge with the capacity for polyolefin biodegradation. The oxidation of polyolefins can be divided into four stages, including biodeterioration, biofragmentation, bioassimilation and mineralization [4]. As shown in Figure 2, (taking PE as an example), degradation begins with the formation of biofilms [3]. In the biodeterioration stage, the surface of polyolefins is initially oxidized by the action of oxidative enzymes released by microorganisms or induced by exterior agents, such as sunlight (ultraviolet) exposure [4,5]. Biodeterioration reduces the number of carbonyl-groups and turns them into carboxylic acids, facilitates the further oxidation of polyolefins. During the biofragmentation stage, the polymer carbon chains are hydrolyzed into fragments with the release of intermediate products, which includes long-chain aliphatic compounds, such as alkanes and alkenes. The enzymes (e.g., laccase, manganese peroxidase and alkane hydroxylase) secreted by microorganisms capable of oxidizing polyolefins are involved in the biofragmentation stage [5]. Small hydrocarbon fragments with 10–50 carbon atoms released by biofragmentation are taken up and metabolized by microorganisms in the bioassimilation stage [21]. The hydrolysis products are transferred within the cell and degraded by the enzymes shown in Figure 2 and converted to microbial biomass with the associated release of carbon dioxide and water in the mineralization stage [4,5]. 

### 2.1. Isolated Microorganisms

Numerous of microorganisms with the capacity for polyolefin degradation, including bacteria, fungi and microbial consortia, have been isolated from the environment, such as soil containing plastic waste, the ocean and the guts of plastic-eating worms. These microorganisms are capable of utilizing polyolefins as sole carbon source or can generate depolymerases involved in polyolefin degradation. Therefore, the screening of these microorganisms is vital for the further degradation of polyolefins. Polyolefin-degrading bacteria can be identified by the following procedure. Soil samples collected from an area containing polyolefin waste was mixed with water and shaken [7]. In order to isolate strains from waxworm guts, worm gust need to be isolated and suspended in water [22]. Then, a portion of the suspension was transferred to polyolefin-containing medium. After incubation for several days, the polyolefin-degrading strains would be isolated through gradient dilution the of medium or collecting the strains on the agar media on which PE fragments were spread [22,23,24]. 

#### 2.1.1. Single Bacteria

At present, most single strains that are capable of degrading polyolefins are bacteria, which can form biofilms on the surface of polyolefins or destroy the surface of polyolefins. Table 1 summarizes some bacteria capable of degrading polyolefins, including *R. ruber* [8,25,26], *Pseudomonas* [11,27,28], *Bacillus* [23,29], *Acinetobacter* [30] etc. *R. ruber* C208, one of the most efficient bacteria for PE biodegradation, is a Gram-positive bacterium isolated from soil and is capable of degrading unpretreated LDPE at a rate of 0.9% per week [7,31]. An increased rate of PE degradation was detected after LDPE was pretreated with UV light [32]. Changes in the molecular weight and molecular number of LDPE were also detected after incubation with *R. ruber* C208 [8]. It was reported that *R. ruber* C208 is also capable of degrading PS, achieving a weight loss of 0.8% after 8 weeks [25]. Moreover, *R. rhodochrous* ATCC 29672 exhibited the ability to degrade PP based on the characterization of changes in the metabolic activity of bacteria, such as ATP content, ADP/ATP ratio and cell viability [33].

As shown in Table 1, in addition to bacteria, fungi have the capacity to degrade polyolefin. Generally, the polyolefin-degrading capacity of fungi is better than that of bacteria, as fungi can generate hydrophobins, which can strengthen the fungal contact with the substrate and enable fungi to use polyolefins as carbon source [34]. *Aspergillus clavatus* JASK1, *Phanerochaete chrysosporium* NCIM 1170, *Engyodontium album* MTP091 and *Curvularia* sp. isolated from landfill soil have also been shown to play an important role in PE biodegradation [35,36,37]. 

Biodegradation should not only be determined by weight loss. Chemical properties, as well as changes in Mw (molecular weight) and Mn (molecular number), which can be characterized Fourier transform infrared spectroscopy (FTIR), X-ray photoelectron spectroscopy (XPS), nuclear magnetic resonance (NMR) and gel permeation chromatography (GPC) analyses, are also vital for the measurement of the degree of polyolefin degradation. After the biodeterioration stage, the nature and occurrence of functional groups on the surface of polyethylene substrates are changed, which can be studied by FTIR [21] spectroscopy, XPS [23] and NMR [38]. After the biofragmentation stage, the length of polyolefins is shorted. The result of biofragmentation can be proven by using GPC to measure changes in Mw and Mn [39]. After the biofragmentation stage, degradation products (such alkanes, alkenes, etc.) can be determined by GC-MS (gas chromatography mass spectrometry) [40]. Moreover, the result of bioassimilation can be determined through measurement of dry biomass weight of polyethylene-containing media. Finally, the percent of mineralization can be analyzed by CO_2_ measurement [28]. Furthermore, scanning electron microscopy (SEM) is another vital method used to characterize the surface features of polyolefins. 

As previously reported, some waxworms can chew and eat plastics [23,41,42,43,44]. *Enterobacter asburiae* YT1 and *Bacillus* sp. YP1 were isolated from the guts of waxworms with the capability of degrading PE. Over 28-day incubation of the two strains on PE films, the physical properties (tensile strength and surface topography), chemical structure (hydrophobicity and appearance of carbonyl groups), Mw (accompanied by the formation of daughter products) and weight loss were detected [23]. Kyaw et al. [11] incubated *Pseudomonas aeruginosa* PAO1 with LDPE films. After exposure to *P. aeruginosa* PAO1, the LDPE sample turned into a mixture of long-chain fatty acids, esters, hydrocarbons, oxygenated chemical compounds predominantly containing aldehydes, ketones, esters and ether groups, unsaturated fatty acids and certain unknown compounds. Given that the structure of polyolefins is similar to that of alkanes, some strains with an ability to degrade alkanes also have an effect on PE degradation. *Alcanivorax borkumensis*, a bacterial strain isolated from the sea that can utilize alkane as carbon source [45] was found to induce a weight loss of 3.5% in 7 days [46].

#### 2.1.2. Microbial Consortia

In addition to single bacteria, there are many microbial consortia that are capable of degrading PE isolated from various environments (Table 2). A mixed microbial consortium consisting of two *Bacillus* sp. and two *Paenibacillus* sp. was isolated from a landfill site. After incubation with the mixed microbial consortium for 30 days, the weight and the mean diameter of the PE sample were reduced 16.7% and 22.8%, respectively [29]. A microbial consortium consisting of *Lysinibacillus xylanilyticus* and *Aspergillus niger* was found to be capable of degrading PE [47]. After UV irradiation, the mineralization percentage increased from 15.8% to 29.5% after incubation for 126 days in soil, and according to FTIR and XRD, the chemical properties also improved. Skariyachan et al. [48] isolated a microbial consortium comprising *Brevibacillus* sps. and *Aneurinibacillus* sp. from waste landfills. During a 140-day incubation of the two strains on PE samples, weight loss for LDPE and HDPE strips reached 58.2% and 46.6% respectively, and weight loss for LDPE and HDPE pellets were 45.7% and 37.2%, respectively. As with a single bacterium, UV pretreatment aided in the degradation of polyolefin samples. In addition, the result of GC-MS analysis indicated that PE samples were degraded into cis-2-chlorovinylacetate, tri-decanoic acid and octadecanoic acid. Muenmee et al. [14] pretreated HDPE, LDPE, PP and PS samples with UV for 200 h, then mixed the plastics together. To simulate a landfill environment, the plastics mixture was placed in simulated lysimeters with a synthetic landfill gas (60% CH_4_:40% CO_2_) and a microbial consortium composed of *Methylocyctis* sp., *Methylocella* sp., *Methylobactor* sp., *Methyloccus capsulatus.*, *Nitrosomonas* sp., *Nitrosomonas europaea, Nitrobacter winogradskyi*, *Nitrobacter hamburgensis*, *Burkholderia* sp., *Pseudomonas* sp. and *Xanthobacter* sp. After a reaction period of 3 months under semi-aerobic landfill conditions where different aeration rates were supplied, the plastic samples were degraded into hydrocarbon and oxygenated compounds, such as aliphatic alkanes, alkenes, alcohols and esters. The degradation products of different plastic types, such as the degraded products of HDPE, are mainly alkanes (C_24_H_50_, C_32_H_66_), alkenes (C_15_H_30_, C_19_H_38_) and some alcohols (C_15_H_32_O, C_20_H_42_O) were found, while for LDPE, only one alcohol (C_11_H_24_O) was found.
microorganisms-10-01537-t001_Table 1Table 1Wild strains capable of polyolefin degradation.Polyolefin Phylum/ClassMicroorganismMicroorganism SourcePretreatmentExperimental ConditionBiodegradation ResultReferencePE**Bacteria***Terrabacteria group/Actinobacteria**R. ruber* C208PE agricultural waste in soilUnpretreated LDPE film Incubation for 8 weeks at 37 °CWeight loss: 7.5%[7]*Terrabacteria group/Actinobacteria**R. ruber* C208PE agricultural waste in soilUV-pretreatedLDPE filmIncubation for 4 weeks at 30 °CWeight loss: 8%[32]*Terrabacteria group/Actinobacteria**R. ruber* C208PE agricultural waste in soilUnpretreated LDPE filmIncubation for 30 days at 30 °CWeight loss: 1.5–2.5%; reduction of 20.0% in Mw and 15.0% in Mn[8]*Terrabacteria group/Actinobacteria**Rhodococcus* sp.Three forest soilsPreoxidized LDPE filmIncubation for 30 days at 25 °CConfirmation of adherence[24]*Terrabacteria group/Firmicutes**Staphylococcus arlettae*Various soilenvironmentsUnpretreated PE film and PE powderIncubation for 30 days at 37 °CWeight loss: 13.6%[22]*Proteobacteria/Gammaproteobacteria**Enterobacter asburiae* YT1 *and Bacillus* sp. YP1Guts of plastic-eating waxwormsUnpretreated LLDPE filmShaken flasks incubated for 60 days at 30 °CWeight losses of 6.1% and 10.7% after incubation with *E. asburiae* YT1 and *Bacillus* sp. YP1, respectively[23]*Proteobacte-ria/Gammaproteobacteria**Stentrophomonas* sp.Plastic debris in soilUnpretreated LDPE filmIncubation for 30 days at 28 °CChange in chemical properties[49]*Proteobacte-ria/Gammaproteobacteria**Stentrophomonas pavanii*Solid waste dumpsiteModified LDPEIncubation for 56 days at 30 °CConfirmed by FTIR[50]*Proteobacte-ria/Gammaproteobacteria**Serratia marcescens*SoilLLDPE powder made of LLDPE filmIncubation for 70 days at 30 °CWeight loss: 36.0%[51]*Proteobacte-ria/Gammaproteobacteria**Alcanivorax borkumensis*Mediterranean SeaUnpretreated LDPE filmIncubation for 7 days at 30 °CWeight loss: 3.5%[46]*Terrabacteria group/Actinobacteria**Streptomyces* spp.Nile River Delta30 °C heat-treated degradable PE filmIncubation for 1 month at 30 °CThree species showedslight weight loss.[52]*Terrabacteria group/Firmicutes**Pseudomonas aeruginosa* PAO1ATCCUnpretreated LDPE filmIncubation for 120 days at 37 °CMaximum weight loss: 20.0%[11]*Terrabacteria group/Firmicute**Dikarya/Ascomycota**Pseudomonas, Bacillus, Brevibacillus, Cellulosimicrobium, Lysinibacillus and Aspergillus*Dump siteUnpretreated PE filmsIncubation for 16 weeks in shaken flasks at 37 °C and 28 °CGravimetric weight reductions of up to 36.4 % and 35.7% recorded for *Aspergillus* sp. and *Bacillus* sp. isolates, respectively.[53]**Fungi***Dikarya/Ascomycota**Aspergillus clavatus* JASK1 Landfill soilUnpretreated LDPE films (bags)Shaken flasks incubated for 90 days Weight loss: 35.0%[35]PP**Bacteria***Proteobacte-ria/Gammaproteobacteria**Stenotrophomonas panacihumi* PA3–2SoilUnpretreated PP powder Incubation for 90 days at 37 °CMw decreased[54]*Terrabacteria group/Actinobacteria**R. rhodochrous* ATCC 29672ATCCPP film with pro-oxidant (Mn, Mn/Fe or Co) additives Incubation for 180 daysChanges in ATP levels[33]*Terrabacteria group/Firmicute**Bacillus flexus*A soil consortium enriched from a plastic dumping siteUV-pretreated PP film Incubation for 1 yearWeight loss: 2.5%[55]*Terrabacteria group/Firmicute**Bacillus cereus*MangrovesedimentsUV-pretreated PP granulesIncubation for 40 days at 3 °CWeight loss: 12.0%[56]
*Sporosarcina**globispora*Mangrove sedimentsUV-pretreated PP granulesincubation for 40 days at 33 °CWeight loss: 11.0%[56]*Terrabacteria group/Firmicute**Bacillus* sp.Municipal compostwasteUnpretreated PP powderIncubation for 15 days at 37 °CWeight loss: 10.0–12.0%[57]**Fungi***Proteobacteria /Gammaproteobacteria**Phanerochaete chrysosporium* NCIM 1170, *Engyodontium album* MTP091
100 °C or UV for 10 daysShaken flasks incubated for 12 monthsWeight loss: 18.8% and 9.4% with *P. chrysosporium* and *E. album*, respectively[37]PS**Bacteria***Terrabacteria group/Firmicute**Exiguobacterium* sp. *strain* YT2Guts of the larvae of Tenebrio molitor LinnaeusUnpretreated styrofoam PS films Incubation for 60 daysWeight loss: 7.4 % Mw decrease: 11.0%[58]*Terrabacteria group/Firmicute**Pseudomonas* sp.SoilUnpretreated high-impact PS filmsIncubation for 30 days at 30 °CWeight loss: more than 10.0% [59]*Terrabacteria group/Firmicute**Bacillus* sp.SoilUnpretreated high-impact PS filmsIncubation for 30 days at 30 °CWeight loss: 23.7%[59]*Terrabacteria group/Firmicute**Pseudomonas**aeruginosa*Degraded polymernanocompositePS: PLA and PS: PLA:organically modified montmorillonite (OMMT) compositesIncubation for 28 days at 30 °C in MSM9.9% degradation at 10 and 25% PS: PLA composites[60]*Terrabacteria group/Firmicute**Pseudomonas**putida* CA-3Industrial bioreactor Pyrolyzed PS48 h offermentation at30 °C, 500 rpmA single pyrolysisrun and fourfermentation runsresulted in theconversion of 64 gof PS to6.4 g of PHA[25]*Terrabacteria group/Firmicute**Exiguobacterium*sp. strain YT2Degraded plasticwasteHigh-impact PSIncubation for 30 days at 30 °CWeight loss: 12.4%[61]*Terrabacteria group/Actinobacteria**R. ruber* C208
Unpretreated styrofoam PS films Incubation for 8 weeks at 28 °CWeight loss: 0.8%[62]**Fungi***Dikarya/Ascomycota**Curvularia* sp.Soil samplesChemically oxidized PSIncubation for 9 weeks at 30 °CMicroscopicexaminationshowed adherenceand penetrance tothe polymer[36]

### 2.2. Engineered Strains

#### 2.2.1. Hydrolases Capable of Polyolefin Degradation

Most the enzymes capable of degrading PE are oxidoreductases. As shown in Table 3, well-known identified enzymes with the ability to oxidize polyolefins include laccase, manganese peroxidase, alkane hydroxylase and soybean peroxidase. Furthermore, a PS-degrading enzyme called hydroquinone peroxidase was identified from *Azotobacter beijerinckii* HM121. However, no PP-degrading enzymes have been identified to date. The result of enzymatic degradation of PE is that oxidation groups are introduced into the PE chain, which means polyolefins cannot be oxidized into monomers as the only act of these enzymes. Most PE-degrading enzymes can only perform terminal oxidation (the terminal carbon in polyolefins be oxidized) and subterminal oxidation (the carbon adjacent to the terminal carbon in polyolefins be oxidized) of PE. For example, laccase and manganese peroxidase can perform terminal oxidation, and the AlkB family can degrade n-alkanes, the main component of polyethylene, through either terminal or subterminal hydroxylation reactions [71]. Therefore, an ideal polyolefin-degrading enzyme has high hydroxylation activity against any carbon in the carbon chain so as to achieve the efficient transformation of polyolefin to its oligomer or monomer. 

Laccases (EC 1.10.3.2), which belong to the so-called blue-copper family of oxidases, can catalyze the oxidation of a wide range of phenols and arylamines. Laccases, which are glycoproteins, have been reported in higher plants, fungi and bacteria. A laccase was purified from *R. ruber* C208 with the ability to degrade LDPE. As laccases contain four copper ion bonding sites, copper markedly affects their induction and activity, resulting in PE degradation. mRNA quantification by RT-PCR revealed a 13-fold increase in laccase mRNA levels in copper-treated cultures compared with an untreated control. The addition of copper to C208 cultures containing PE enhanced the biodegradation of PE by 75% [7,72]. A laccase mediator system (LMS) is composed of laccase and some small-molecule compounds that are easily oxidized by laccase, such as HBT, ABTS and DMP. In the process of LMS oxidation, laccase oxidizes the mediator first; then, the oxidized mediator oxidizes the substrate. It was reported that in the presence of a mediator, laccase can oxidize some substrates that it cannot oxidize alone. HBT, which has been used for PE degradation, reacts with non-phenolic models by a radical mechanism involving hydrogen atom abstraction [73]. Some researchers treated polyethylene with LMS using HBT (0.2 mM) as a mediator; after 3 days, the polyethylene membrane exhibited no elongation, and its relative tensile strength decreased by about 60%, which is higher than in the absence of HBT (20%) [74]. Johnnie et al. used laccase from *Trichoderma viride* fungus and 1-HBT to degrade LDPE. After incubation for 10 days, the weight loss of LDPE came to 2.3% [75]. 

Cytochrome P450 (CYP, P450), a member of a superfamily of heme–thiolate proteins, is distributed in most living organisms. There are more than 300,000 P450 genes. However, no P450 genes were found in *E. coli*, which means that *E. coil* is a good chassis for the heterologous expression of P450 genes. P450 enzymes can identify multiple substrates and catalyze diverse reactions, such as C–H hydroxylation; C=C double-bond epoxidation; heteroatom oxygenation; O-, N- and S-dealkylation; aromatic coupling; and C–C bond cleavage [76]. Because other PE-degrading enzymes can only perform terminal oxidation and subterminal oxidation of PE, the application of an ideal P450 enzyme that can cleave PE into short chains would contribute to the biodegradation of PE. 

Alkane, which is composed of C–C bonds and C–H bonds, has a similar structure to that of PE. Therefore, alkane mono-oxygenase enzymes are potential candidates for the degradation of PE. One alkane hydroxylase, namely AlkB, has been reported to degrade PE [76]. AlkB, which was first identified in alkane-consuming *Pseudomonas* species isolated from oil-contaminated areas, is a membrane-bound, non-heme di-iron monooxygenase [77]. 

Manganese peroxidase was purified from a lignin-degrading fungus: *P. chrysosporium*. Manganese peroxidase was first identified as a lignin-degrading enzyme. It was reported that the addition of Mn (II) to nitrogen- or carbon-limited culture medium enhanced PE degradation [9]. 

#### 2.2.2. Engineered Chassis for Polyolefin Biodegradation

As shown in Table 3, the microorganisms that are capable of secreting polyolefin biodegrading enzymes are not model organisms, which means that they are difficult to genetically engineer. Two model organisms, *E. coli* and *Y. lipolytica,* have been applied to the expression and secretion of polyolefin-biodegrading enzymes to date [12,78,79]. The heterologous expression of polyolefin-degrading enzymes in model organisms can efficiently increase the expression level of polyolefin-degrading enzymes through genetic engineering in model organisms.

***E. coli*:***E. coli* is one of the most widely used model microorganisms for production of recombinant proteins. As a model microorganism, *E. coli* has advantages in many aspects, such as a simple genetic background, ease of genetic modification and simple growth conditions. Engineered *E. coli* has been used to express alkane hydroxylase to degrade LMWPE. The *alkB* gene, which was cloned from *Pseudomonas* sp. E4, was introduced into *E. coli* BL21. After incubation for 80 days at 37 °C with engineered *E. coli,* 19.3% of the LMWPE was degraded [28]. The rubredoxin and rubredoxin reductase could help alkane monooxygenase to transfer electron. If the rubredoxin and rubredoxin reductase are co-expressed with alkane monooxygenase, the conversion rate can be increased. Researchers fused and expressed *alkB* with its coenzyme genes—*rubA1, rubA2 and rub*—in *E. coli* BL21. The result indicated that 30.5% of the carbon of LMWPE-1 degraded into CO_2_ after 78 days [80]. Another study revealed that an alkane-1-monooxygenase (AlkB) in *Acinetobacter johnsonii* JNU01 degraded PS, and this finding was later confirmed by recombinant *alkB* in *E. coli* BL21 [53]. These researchers also expressed *alkB2* in *E. coli,* and the result indicates that *alkB2* was more efficient for low-molecular-weight PE biodegradation than *alkB1* [12]. A laccase gene isolated from a marine fungus was expressed in *E. coli* and showed PE degradation ability [81].

Many recent studies have shown that engineered *E. coli* can be used for laccase heterologous expression and secretion [82,83,84]. Ihssen et al. [78] expressed five novel bacterial laccase-like multicopper oxidases (LMCOs) of diverse origin. However, a potential issue with laccase expression in *E. coli* is that it is easy for *E. coli* to form inclusion bodies when expressing extracellular enzymes. Mo et al. [79] expressed three laccases from three different organisms, namely Lac1326 from marine sediment samples, fungal tvel5 laccase from *Trametes versicolor* and bacterial BPUL laccases from *Bacillus pumilus* for the purpose of degrading β-estradiol. The result of Western blot analysis indicates that laccase was detected both in vivo and in vitro in *E. coli,* which means that some laccase stayed in inclusion bodies instead of being secreted in vitro. Given that laccase has been widely used for the biodegradation of PE [38,75,85,86], the heterologous expression of laccase in *E. coli* is a potential method for the biodegradation of PE. 

***Yarrowia lipolytica*:***Y. lipolytica* is a Crabtree-negative ascomycete yeast with good protein secretion capacities. Compared to other yeasts, *Y. lipolytica* lacks α-1,3-mannosyltransferase, a factor that limits the amount of excessive mannosylation of secreted heterologous glycoproteins and constitutes a valuable asset for the production of therapeutic proteins [87]. *Y. lipolytica* W29 is a wild-type strain with a remarkable characteristically high secretion level of proteins [88]. *Y. lipolytica* W29 ura302 was obtained through genetic convention of URA3 into ura3-302 in *Y. lipolytica* W29, and it was able to utilize sucrose and molasses as a carbon source under the control of XPR2 promoter. After the genetic convention of XPR2 into xpr2-322 and AXP1 into axp1-2, which indicates inactivation of alkaline extracellular protease and acid extracellular protease that would degrade foreign extracellular protein, a strain with high heterologous protein production capability called *Y. lipolytica* Po1f was obtained. A new strain called *Y. lipolytica* Po1g that carries a pBR322 docking platform was obtained through the integration of PINA300′ plasmid in *Y. lipolytica* Po1f [89]. *Y. lipolytica* Po1g was induced with a YLEX kit for expression/secretion of heterologous proteins [90]. A laccase from the white-rot fungus *Trametes versicolor* was expressed in *Y. lipolytica* Po1g for the biodegradation of PE. Compared to the yeast secretion signal, the native secretion signal showed higher enzyme activity in the culture medium. The yield of laccase reached 2.5 mg/L (0.23 units/mL) [72]. Laccase has been widely used for the biodegradation of PE, and the heterologous expression of laccase in *Y. lipolytica* is a promising method of PE degradation.

**Table 3 microorganisms-10-01537-t003:** Enzymes capable of polyolefin degradation.

Plastics	Enzymes	Enzyme Source	Pretreatment	Experimental Condition	Result	Reference
PE	Laccase	*R. ruber C208*	Unpretreated LDPE film	Incubation for 30 days at 30 °C	Weight loss: 1.5–2.5%; reduction of 20% in Mw and 15% in Mn	[7]
	Manganese peroxidase	*Phanerochaete chrysosporium*	Unpretreated PE film	Incubation for 12 days at 37 °C	Mw decreased	[9]
	Soybean peroxidase	Soybean	Unpretreated HDPE film	Reaction for 2 h at 60 °C	Hydrophilicity increased	[10]
	Alkane hydroxylase	*Pseudomonas* sp. E4	Unpretreated LMWPE sheet	Incubation for 80 days at 37 °C	Weight loss: 19.3%	[28]
	Alkane hydroxylase	*Pseudomonas aeruginosa* E7(uniport Q9I0R2)	Unpretreated LMWPE film	Incubation for 50 days at 37 °C	Weight loss: 19.6–30.5%	[12]
PS	Hydroquinone peroxidase	*Azotobacter beijerinckii* HM121	Unpretreated PS film	Incubation for 20 min	Mw decreased	[13]
	Alkane hydroxylase	*A. johnsonii* JNU01	Unpretreated low-molecular-weight PS powder	Incubation for 7 days at 28 °C	Confirmed by FTIR and SEM	[72]

## 3. Artificial Microbial Consortia in Polyolefin Biodegradation

Microorganisms do not exist independently in natural environments, and they usually live in complex communities. They communicate through quorum sensing and share metabolites and enzymes, expanding the substrate range compared with single bacteria. Therefore, artificial microbial consortia can be constructed to degrade plastics that are not easily degraded naturally [91]. The application of artificial microbial consortia for complex biological processes is an emerging field in synthetic biology. There are several specific advantages of plastic degradation by artificial microbial consortia compared to pure culture. Firstly, the synergies of different strains and different enzymic systems can improve the efficiency of polyolefin degradation and reduce the metabolic burden [92,93]. Moreover, the effect of polyolefin degradation by artificial microbial consortia can be more complete than that of pure culture. Finally, the construction of artificial microbial consortia is a time-saving and efficient method relative to other metabolic engineering techniques [79]. Therefore, constructing artificial microbial consortia is regarded as a promising means of polyolefin degradation. The bioproduction of surfactin by coculture of *B. amyloliquefaciens* MT45 and *B. amyloliquefaciens* X82 improved substrate utilization and increased the product titer by 3.3-fold [94]. Another microbial consortium including a crude oil degrader and biosurfactant producer was constructed to degrade crude oil, achieving 95.8% degradation efficiency of crude oil and degrading various hydrocarbons more effectively than single strains [95]. In our laboratory, we constructed a four-microbe consortium comprising two metabolically engineered *B. subtilis*, *Rhodococcus jostii* and *P. putida* to degrade PET. The result showed that the microbial consortium could degrade PET film, with weight loss reaching 23.2% under ambient temperature. The artificial microbial consortium successfully relieved the metabolic inhibition of TPA and EG [96,97]. Some researchers also proposed a novel strategy combining enzymic and microbial degradation of PET, achieving a maximum degradation efficiency of approximately 91.4% [98]. As for PET degradation, artificial microbial consortia provide a potential means of polyolefin degradation.

Most microbial consortia with the ability to degrade polyolefins identified to date are natural microbial consortia. Skariyachan et al. [66] formulated a bacterial consortium composed of strains isolated from plastic garbage processing areas. After incubation with the bacterial consortium, LDPE strips and LDPE pellets showed weight loss of 81.0% and 38.0%, respectively. Another artificially formulated microbial consortium composed of four strains isolated from cow dung samples gathered from highly plastic-acclimated environments degraded 75.0%, 55.0%, 60.0% and 43.0% of LDPE strips, LDPE pellets, HDPE strips and HDPE pellets, respectively, over a period of 120 days at 55 °C. The weight loss of PE degraded by microbial consortia is much higher than that of pure culture (10.2% at most) [65]. Later, these researchers formulated a bacterial consortium from cow dung samples for the purpose of LDPE and PP degradation. After incubation with the potential consortium (CB3) at 37 °C, LDPE and PP films showed degradation of 64.2% and 63.0%, respectively, which is much higher than that of pure culture of any strains in this microbial consortium [99]. However, artificially formulated microbial consortia still have disadvantages, such as division of labor, metabolic imbalance, competition, cooperation and complex interactions [100].

Most single bacteria or microbial consortia isolated from nature can only degrade polyolefins into long-chain aliphatic compounds (alkanes, alkenes, ketones, aldehydes, alcohols, acid, ketone acids, dicarboxylic acids and esters) at most. Few studies have achieved the goal of complete degradation of polyolefins. Building artificial microbial consortia provides a potential means for the complete degradation of polyolefins using polymer-degrading strains and long-chain aliphatic-compound-degrading strains. Combining polyolefin-degrading modules and long-chain aliphatic-compound-degrading module is promising means by which to completely degrade polyolefins (Figure 3). As shown in Figure 3 (taking PE as an example), an artificial microbial consortium that includes a biosurfactant producer [101], a polyolefin degrader [72], an alkene degrader and an alkane degrader [102] could be constructed for the biodegradation of polyolefins. In this artificial microbial consortium, the biosurfactant producer could produce biosurfactant, which can help the polyolefin degrader or polyolefin-degrading enzymes to make contact with polyolefins [15]. Then, some carbonyl groups in the carbon chain of polyolefin would be reduced under the action of the polyolefin-degrader or extracellular enzymes secreted by engineering strains. Subsequently, the polymer carbon chains would be hydrolyzed into fragments with the release of intermediate products, including alkane, alkene, ketones, aldehydes, alcohols, acids, etc. These intermediate products could be transported to alkene degraders and alkane degraders [21]. Alkanes could be gradually degraded into ketones, aldehydes, alcohols and acid and enter the tricarboxylic acid cycle (TCA cycle). As for alkenes, they could be degraded into epoxide, vicinal diol, aldehydes and acids and enter the metabolic pathway in the cell [5,76].

In contrast to polyolefins, long-chain aliphatic compounds can transfer into cells through active transport, passive transport, endocytosis and free diffusion and be metabolized in the cells. Many researchers have reported microorganisms isolated from environments that are capable of degrading alkanes, such as *Aspergillus* sp. [103], *Bacillus* sp. [104] and *Yarrowia* sp. [105]. Many enzymes have been identified for alkane degradation, such as alkane hydroxylase, which can oxidize alkanes into alcohol. Alcohols are then gradually oxidized into aldehydes, acids and acetyl-CoA and enter the tricarboxylic acid cycle (TCA cycle) or are converted into high-value chemicals, such as sophorolipid [101]. As for the degradation of alkene, the biodegrading enzymes are alkene monooxygenase, which can oxidize alkenes into epoxide [17]. Alkenes could be oxidized into vicinal diol [106] and then be oxidized into aldehydes, acids and acetyl-CoA and enter the TCA cycle.

Constructing an artificial microbial consortium that includes part or all of the following strains is a vital metabolic strategy to solve the current problems of polyolefin degradation: polyolefin degraders, long-chain aliphatic-compound degraders and biosurfactant producers. Artificial microbial consortia can perform more complex tasks compared to pure strains, and the division of labor is clearer than that of artificially formulated consortia.

## 4. Conclusions

In this review, we summarized the current advances in polyolefin biodegradation from pure natural strains, natural microbial consortia, enzymes, engineered chassis and artificial microbial consortia. The prospect of the biodegradation of polyolefins by artificial microbial consortia was also discussed. Constructing artificial microbial consortia is a promising strategy for the biodegradation of polyolefin. The use of artificial microbial consortia is a promising method for the degradation of polyolefins and long-chain aliphatic compounds. Compared to natural microbial consortia, the division of labor in artificial microbial consortia is clearer and more favorable for follow-up studies. The possibility of building an artificial microbial consortium including a biosurfactant producer, a polyolefin degrader, an alkene degrader and an alkane degrader was proposed in this review. There have been many recent advances in synthetic biology and metabolic engineering [107,108,109,110,111]. Therefore, it is possible to design rational and efficient artificial microbial consortia to degrade polyolefins, as well as more complex compounds.

## Figures and Tables

**Figure 1 microorganisms-10-01537-f001:**
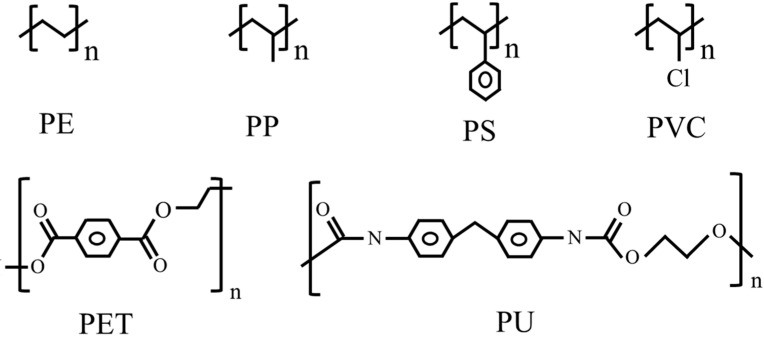
Structures of major synthetic polymers.

**Figure 2 microorganisms-10-01537-f002:**
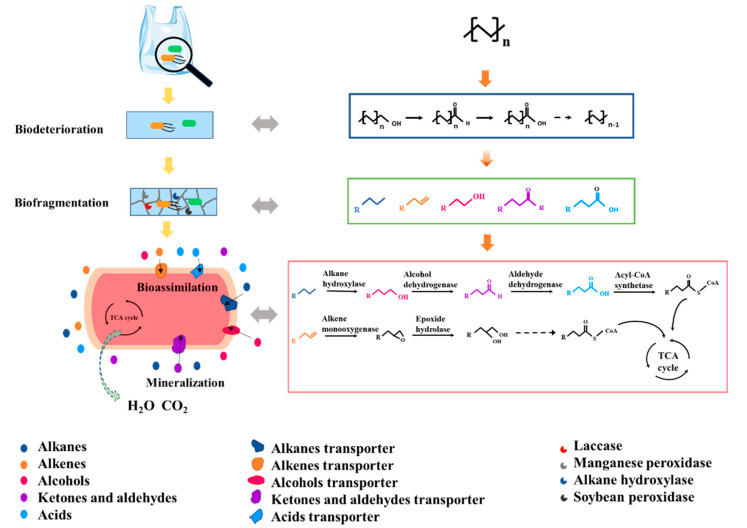
The process of PE biodegradation.

**Figure 3 microorganisms-10-01537-f003:**
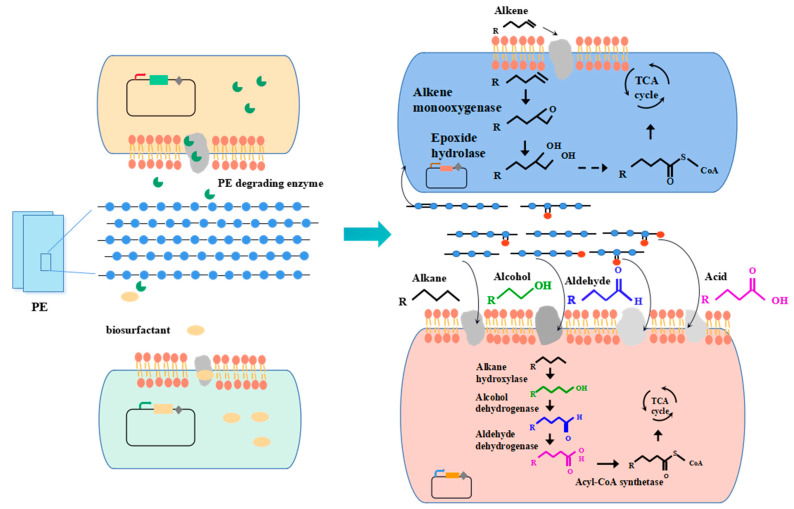
Artificial microbial consortia for complete PE biodegradation.

**Table 2 microorganisms-10-01537-t002:** Microbial consortia capable of polyolefin degradation.

Polyolefin	Microorganisms	Microorganism Source	Pretreatment	Experimental Condition	Biodegradation Result	Reference
PE	Mixed microorganisms	Microbial activated soil	Thermally pretreated	Incubation for 180 days at 60 °C	Mineralization percentage: 60.0%	[26]
	*Lysinibacillus xylanilyticus* and *Aspergillus niger*	Landfill soils	UV-irradiated and non-UV-irradiated LDPE films	Incubation for 126 days in soil	Mineralization percentage: 29.5% (UV-irradiated)	[29]
	Soil microorgannisms	Soil	UV-irradiated and non-UV-irradiated	Incubation for 28 days	Weight loss: 6.0% and 3.5%	[63]
	*Comamonas, Delftia*, and *Stenotrophomonas*	Degraded plastic debris	Unpretreated LDPE films	Shaken flasks incubated for 90 days at 28 °C	Changes in chemical properties	[64]
	*Brevibacillus* sp. and *Aneurinibacillus* sp.	Waste management landfills and sewage treatment plants	Unpretreated HDPE, LDPE films and pellets	Incubation for 140 days at 50 °C	Weight loss for LDPE and HDPE strips was 58.2% and 46.6% respectively; weight loss for LDPE and HDPE pellets was 45.7 % and 37.2%, respectively	[47]
	*Bacillus* sp. and * Paenibacillus* sp.	Landfill site	Unpretreated PE microplastic granules	Incubation for 60 days at 30 °C	Weight loss: 16.7%; mean diameter reduction: 22.8%	[46]
	Artificial thermophilic bacterial consortium composed of bacterial isolates *(Bacillus vallismortis, Pseudomonas protegens, Stenotrophomonas* sp. and *Paenibacillus* sp.)	Dung of cows fed off plastic-contaminated pastures	LDPE and HDPE films and pellets	Incubation for 120 days at 55 °C	Gravimetric weight loss percentages of 75.0%, 55.0%, 60.0% and 43.0% for LDPE film, pellets, HDPE film and pellets, respectively	[65]
	Two *Enterobacter* sp. and one *Pantoea* sp.	Plastic garbage processing areas	LDPE films and pellets	Incubation for 120 days at 37 °C	Maximum weight loss: 81.0%	[66]
	*Lysinibacillus.* sp. and *Salinibacterium* sp.	Plastic samples and surface water	LDPE and HDPE pieces	Incubation at 25 °C for 6 months	Weight loss: 15.0% for LDPE after 4 months and 5.5% for HDPE after 6 months	[67]
pp	*Bacillus* and *Pseudomonas*		UV- or thermally pretreated PP films	Flasks incubated at 28 ± 2 °C and 180 rpm for 12 months	Weight loss: 1.9%	[68]
	Microbial consortium	Plastic dumping site	Thermally pretreated PP films	Incubation for 1 year	Weight loss: 10.7%	[69]
	Mixed soil community	Soil samples rich in plastic waste	Isotactic PP films	Incubation for 5 months	The film had 40% methylene chloride extractable compounds, and a mixture of hydrocarbons (between C_10_H_22_ and C_31_H_64_) was detected and identified in the extract	[70]

## Data Availability

Not applicable.

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
