# Peer review of "Current Advances in Biodegradation of Polyolefins"

_microorganisms, 2022, doi:10.3390/microorganisms10081537_

Round 1
Reviewer 1 Report
In general, the review article was focused on recent publications on bacteria and fungi and their enzymes involved in the biodegradation of polyolefins. The authors highlighted the use of microbial consortia and genetically engineered organisms. The paper was clearly written. It would have been useful but not essential, if the authors mentioned other review articles relevant to this subject. There were several grammatical corrections made on the attached pdf of the authors manuscript. There are no scientific or technical concerns.

Reviewer 2 Report
Overall assessment:
In this manuscript, the authors reviewed the progress on polyolefins biodegradation by single strains, microbial consortia, and enzymes. The authors have summarized the main results of well-studied strains, which have been isolated from natural sources, and also discussed the construction of artificial consortia that can help enhancing polyolefins biodegradation, (mostly) in term of weight loss of the plastics. The topic of this review paper is of interest, since most of the studies focus on the easier degradable polyesters. I would however recommend improving the structure and carefully revising the language throughout the text. This includes a more precise and careful use of scientific terms and avoiding too generic discussions (but provide concrete examples instead). English grammar, syntax, and proof reading of sentences (sometimes incomplete) need to be improved substantially. In general, a more in depth and comprehensive presentation of the topics would be beneficial. The author could further improve the manuscript by mentioning the importance of using biodegradation, the advantages and prospects of integrating biodegradation into current polyolefins waste management, and clearly stating why this review manuscript is needed. I provided here a few suggestions that could help authors improve the manuscript.
General comments:
GC1: There are several errors regarding writing of scientific name of bacterial strains. Some of them were correctly written in italic, but some others were not (For example, in between line 85-93, in Table 2, line 163, 182, 200, etc.). Please check and fix the errors.
GC2: The different typing style of “°C” is detected all over the manuscript. Please check and provide homogenous style.
GC3: Check the misspelling words; for instance, in Table 2 “persentages” and Line 193 “non-haem”.
GC4: Check the writing with subscription, e.g., CO2.
GC5: The introduction is rather simplistic. A more comprehensive background of the problem could have been provided, including a synthetic overview of the current recycling technology, sources and applications of polyolefins on the market, faith of end-of life, challenges in current management of plastic waste and why/how/in what cases biodegradation can contribute, etc.
GC6: Improve the conclusion by mentioning the most significant results so far and future prospective of proposed idea.
Specific comments:
- Line 37: I would argue that this depends on the plastic type. The opposite was found with PET for instance. Can you be more specific?
- Line 41: please correct “strained” into strain.
- Line 45: I am wondering if it is correct to state that these enzymes are really capable of degrading PE? Or is it rather that can they contribute to a particular step of PE degradation? Maybe you can elaborate and be more precise?
- Line 48: please reformulate this sentence
- Line 65: have there been reports of PE mineralization (complete degradation to CO2 and H20)? What metabolic pathways are used?
- Lines 66-67: I perfectly agree that physical parameters are very important and contribute to the plastic deterioration, but then we cannot call it BIOdeterioration. A more careful use of terms would be important.
-Line 65: I am not sure i follow the "reduction of number of carbonyl groups"? Does the oxidation reduce the number of carbonyl groups in polyolefins?
- Lines 71-72: maybe also here be more precise, else it sounds like laccases are involved in the hydrolysis of the polyolefins rather than the oxidation.
- Lines 74-75: also here the authors could be more specific: are all hydrolysis products able to be transferred inside the cells? If not, which ones are and which ones are not? And how are they converted into microbial biomass?
- Paragraph 2.1: I would recommend to re-check the English here. For instance, what are “depolymerases involved in polyolefin”? Also, maybe the authors could elaborate a bit with a paragraph dedicated to the different screening methods (classical and more advanced).
-Lines 88-89: this is another incomplete/truncated sentence that is missing a conclusion. Please proof-read all your text, else it feels like a draft.
- Figure 1: this figure is an interesting attempt to make a schematic of this complex process. Might be improved but is definitely a good idea. For instance: looking at the right side of the figure, under the biofragmentation step: if these are chemical formulas representing monomers then this would not be biofragmentation but already assimilation? Or does biofragmentation produce all monomers (so no fragments)? Regarding the bottom right (pink square): maybe you could make the letters and numbers a bit bigger so they can be read?
- At line 109-112, you mentioned the other methods (e.g., FTIR, XPS, NMR, GPC) analyzing chemical properties of polyolefins could help determining the biodegradation. Please provide the description on how they help illustrate the biodegradation or give examples/ include some previous results from the analysis by those methods.
- Lines 122-124: Accurate measurements to characterize polyolefin biodegradation is rather challenging and a key issue in such studies. I would recommend the authors to add a paragraph dedicated to the most important methods used to characterize polyolefin degradation and creation of oligomers and metabolites.
- Paragraph 2.1.2: English needs proper revision and proof reading by a native speaker
- Lines 131-132: a mixed consortium isolated from a landfill that is made of 2 species only? how was it obtained?
- Lines 134: I would reformulate this sentence. “isolate” a consortium sounds odd and “is consist” is grammatically wrong.
- Lines 141-142: i suggest to reformulate the whole sentence
- Line 144: where did the 2-cholorovinylacetate come from, if the starting material was PE? There should be no chlorine if it is a pure polymer?
- Line 146: Can you explain better the experimental conditions used? Furthermore, what is a “simulated lysemeter” and why did they researchers add methane to degrade polyolefins?
- Paragraph 2.2.1: the paragraph could be improved. Only relatively few well-known enzymes are actually discussed and presented and listed in a non structured way (feels like a check list rather than a coherent discussion). Polyolefin depolymerization is not just about laccase, monooxigenase and manganese peroxidase. These are the most reported enzymes but a more in depth analysis of literature should be presented here. Would the oxidation of polyolefins be enough to obtain monomers?
- Line 164-166: this sentence seems to be somewhat incomplete and grammatically incorrect. Please reformulate.
- Lines 170-171: please double check the sentences here.
- Line 178: this is an interesting topic but a rather generic and vague discussion of the importance of mediator. Please add examples of mechanism and type of mediators that have been investigated and the effect on the polyolefin enzymes.
- Lines 186-188: This sentence is not really understandable and seems out of context. You have not introduced the concept of terminal and sub-terminal oxidation of polyolefins. So you could first explain which enzymes are responsible for what (terminal or sub-terminal oxidation) and then explain how P450 can help further.
- Lines 189-192: from what I can see, you have a separate paragraph 2.2.2 about engineered chassis but already mentioned 2 examples here. I suggest you revise the structure of your MS a bit more carefully.
- Lines 194-195: I am not sure i follow. Can you elaborate a bit more the research conditions presented here? Heterologous expression of AlkB in E.coli allows for complete mineralization of 20% PE to CO2? The enzyme (alone) should contribute to the oxidation step but how can it lead to mineralization to CO2 without whole-cell metabolism?
- Lines 195-199: I think this is an interesting topic. Please elaborate and explain how rubredoxin and rubredoxin reductase contribute to the increased conversion rate. How do they interact with the enzymatic process? Furthermore, to improve the scientific value of this MS i recommend you do not just present a list of different findings from literature but put them in a broader context, helping the reader understanding why these findings are relevant for this topic, what are the underlying mechanisms that explain certain observations, and how these topics are connected together.
- Lines 200: please revise the sentence. Else it seems like manganese peroxidase is a fungus?
- Paragraph 2.2.2: Can you elaborate on why it could be relevant to use heterologously expressed enzymes instead of the mixed cultures you presented in the previous paragraph? To my knowledge, in real life the bottleneck of this approach, in the case of plastic degrading enzymes, is often exactly the difficulty of reaching a proper expression level (not to mention secretion).
So in many cases it is not that simple to overexpress these types of enzymes.
- Lines 219-222: were these examples leading to a more efficient degradation than when using the wild type strains that naturally secrete these enzymes? What were the main challenges of these engineered systems compared to the state of the art of heterologous expression for the more investigated polyester (PET) degradation?
- Lines 251-252: I am not sure I understand. Can you elaborate or explain more? How is the secretion signal related to the enzyme activity?
- Line 254: this is a rather generic statement, which should be supported by concrete examples from literature, showing improved yields and/or enzyme activities for PE degradation (compared to wild type performance).
- lines 260-262: I perfectly agree but then you would better introduce this concept of synergies among strains and enzymes in the respective paragraphs that deal with strains and enzymes. This is a key point in polyolefin degradation (compared for instance to polyesters) so concrete examples of microbial and enzymatic synergies for polyolefin degradation should be presented. Or at least add a reference here, because you have not really discussed this topic so far.
_ lines 263-264: Please reformulate the sentence
- Lines 268-270: I would argue that this is also true in the case of adapted mixed consortia. This is exactly the point of using mixed consortia that can exploit metabolic collaboration, cross-feeding mechanisms, etc. I would suggest that you first introduce this concept before discussing the importance of the artificial consortia. You can find some additional information about designing artificial consortia for plastic degradation in literature (for example: Microbial Degradation of Plastics: New Plastic Degraders, Mixed Cultures and Engineering Strategies by S. Jenkins, A. Martínez i Quer,C. Fonseca, C. Varrone. Book Editor(s):Nazia Jamil,Prasun Kumar,Rida Batool; 2019. https://doi.org/10.1002/9781119592129.ch12)
- Line 280: this sentence is not understandable and should be reformulated
- Reference is missing from the following sentences.
· Lines 305-307, “In this artificial microbial consortium, the 305 biosurfactant producer could produce biosurfactant, which can help the polyolefins de-306 grader or polyolefins degrading enzymes to contact with polyolefins.”
· Line 315-317, “Alcohols are then gradually oxidized into aldehydes, acids and acetyl-CoA, and enter the tricarboxylic acid cycle (TCA cycle) or be converted into high value chemicals, such as sophorolipid.”
- In Table 1 and 2, the header should be provided between the new sections (e.g., between PE and PP) to prevent the loss of track of what does each column representing.
- In Table 1 at PS section, there is inconsistency in using capital and non-capital letters (high impact PS films -vs- High Impact PS films).
- In Table 2, the results from reference [54] “gravimetric weight reductions of up to 36.4 % and 35.7% recorded for Aspergillus sp. and Bacillus sp. isolates, respectively.” does not seem to be to refer to consortia degradation. Please recheck the data.
- In Table 2 at PP section, authors mentioned PP is degraded by “microbial consortia” (reference [59]). Did the reference mention the members of consortia, or was it open consortia?
- Line 291-294, from the sentence is not clear what the authors wanted to communicate. What is the difference between artificial consortia and artificial formulated consortia? And why is division of labor and cooperation a disadvantage?
I would once again recommend that you present a subparagraph or section dedicated to the use of mixed consortia (where advantages and disadvantages are briefly presented). Else please add some specific references here.
- Line 295: what does “isolated from natural” mean?
- Lines 295-309, authors mentioned that the biosurfactant producer can help increase polyolefins degradation. Please add some literatures supporting the idea.
- Line 310: are you sure that long-chain compounds can transfer into the cell? Can you provide a reference? Or do you mean the short-chain compounds maybe?
- Figure 2: Figure and caption (explaining the different symbols) need to be improved. It is not sufficiently clear if the long-chain PE oligomers get inside the blue cell or are linked to the alkenes? In the pink cell it is not clear whether the transformations from alkane to alcohol, aldehyde and acid take place intra or extracellularly. The enzymes are reported inside the cells but the chemicals are both inside and outside.
Please double-check that all the mentioned enzymes are intracellular as shown in the figure (right part)
-Line 325: Could you please indicate concrete examples of polyolefin degrades, long-chain aliphatic degraders and biosurfactant producers that could be used, based on the literature?
Reviewer 3 Report
The manuscript presented here, titled Current advances in Biodegradation of Polyolefins, provides a very interesting approach to the state of the art in the biodegradation of these polymers. Authors highlight boldly here the promissing capabilities of engineered microbial strains and microbial consortia for the future research on plastics biodegradation. However, the lack of some figures and references produce the need to improve this manuscript.
Firstly, a deeper explanation about the difference among Polyolefins and other polymers, as well as PE from HDPE, LDPE, LLDPE, PP and PS, would be necessary. Beyond to say that some are more difficult to degrade than others. A graphical scheme showing the chemical differences would help to show this.
Since authors are mentioning different types or organisms degradation these compounds, a phylogenetic tree or similar figure showing at which taxonomic groups (domains, phyla, etc) the capability to degradade polyolefins has been observed would be very interesting to check visually. As suggestion, references could be added inside this figure. This would be better than to check the table attached and reduce the tables to just the one regarding consortia (table 2).
For the reader of the manuscript, a straight comparative about the differences on biodegradation rates among single microorganisms, consortia and engineered strains could make clearer the improvements of using cosortia or engineered strains over just single bacteria. A comparison showing the differences on degradation time, environmental temperature at degradation was performed would be interesting to check visually. At least using the weight loss as main unit of degradation, since is the main information provided on the references mentioned. Is the biodegradation efficiency equal at different temperature? Is the biodegradation rate of consortia better than single microorganims at different temperatures?
How exactly PE is degraded by this enzymes/microorganisms/consortia? How the hydrolases mentioned attack the PE molecules?
Figures need further explanation.
Genera and species names must be on italics.
There are some paragraphs without or very few references (i.e. from line 66 to 76, from line 295 to 309) and some sentences that would need also references (line 114, mentioning the waxworms eating plastics; line 338, about the advantages on synthetic biology and metabolic engineering).
Reviewer 4 Report
This review article summarizes the current knowledge on the biodegradation of polyolefins. The contents and the manuscript organization seem almost sufficient. But, there are many English errors; thus, revisions and corrections are needed especially on English writing and terminologies throughout of the manuscript.
1. The contents in L102-107 are confusing because section 2.1.1 is set for single bacterium, not fungi.
2. L130-131: The reviewer wonders and is confused if the consortium is consisted of only two bacteria?
3. Are the sentences in L266-271 related to the degradation of polyorefins?
4. L278: 91.4% is not the degradation rate, but the degradation ratio/efficiency. “Rate” is the speed, thus includes the dimension of time. To avoid misunderstanding, correct terminology is needed.
5. Figure 2 is too complex without sufficient explanation, thus difficult to understand what it indicates. Some explanations are made in L302-309, those in L305-307 are not clearly linked to the figure. The figure and explanations must be improved.
6. English problems are as follows. In addition to the corrections of the followings, English editing by native speakers throughout of the manuscript is necessary.
Whole parts: Genus and species names must be written in italic form. “sp.” must ‘not’ be written in italic form. These general rules are not followed in main text and Tables.
Whole parts: Single bacterium are ‘isolated’, but microbial consortium/consortia are ‘not’ isolated.
L43: secret is wrong as the verb of this sentence. Also, in L204, secret should be secreting.
L63: Remove the period after Figure 1.
L67: Remove the comma after sunlight.
L110: Define the abbreviations, Mw and Mn. The same for GC-MS in L143.
L111: Change Fourier to fourier.
L138-139: Change waste management landfills to waste landfills.
L153: Change degrade products to degradation products.
L172-173: Remove commas after RT-PCR and levels.
L176: What is prosses?
L178: Change present to presence.
L182: founded should be found.
L195 and other parts: 2 in CO2 must be subscript.
L197-198: The uses of “find” would be wrong grammatically.
L222: Change is to was.
L280: Rewrite “most microbial that with”.
L282: Finish a sentence at “area” and start next sentence from “After”.
L287: Remove (p < 0.05).
L295: Correct “from natural”.
L297-298: Correct “Few research has”.
